# Reliability and Validity of the Japanese Version of the Assessment of Readiness for Mobility Transition (ARMT-J) for Japanese Elderly

**DOI:** 10.3390/ijerph192113957

**Published:** 2022-10-27

**Authors:** Satonori Nasu, Yu Ishibashi, Junichi Ikuta, Shingo Yamane, Ryuji Kobayashi

**Affiliations:** 1Department of Occupational Therapy, Graduate School of Human Health Sciences, Tokyo Metropolitan University, Tokyo 116-8551, Japan; 2Department of Occupational Therapy, Nakaizu Rehabilitation Center, Shizuoka 410-2507, Japan; 3Department of Occupational Therapy, Faculty of Health Sciences, Aino University, Osaka 567-0012, Japan; 4Department of Occupational Therapy, Faculty of Health Sciences, Okayama Healthcare Professional University, Okayama 700-0913, Japan

**Keywords:** driving cessation, community mobility, older adults, assessment

## Abstract

The Assessment of Readiness for Mobility Transition (ARMT) questionnaire assesses individuals’ emotional and attitudinal readiness related to mobility as they age. This study aimed to examine the reliability and validity of the Japanese version of the ARMT (ARMT-J). The ARMT-J and related variables were administered to 173 patients and staff members undergoing rehabilitation at hospitals in Japan. Construct validity was first examined using confirmatory factor analysis (CFA) to confirm cross-cultural validity. For structural validity, the optimal number of factors was confirmed using a Velicer’s minimum average partial test and parallel analysis, followed by exploratory factor analysis (EFA). Finally, a CFA was performed using the most appropriate model. Internal consistency, test–retest reliability, standard error of measurement (SEM), and smallest detectable change (SDC) were assessed for reliability. The CFA fit for the factor structure of the original ARMT was low. Therefore, the EFA was conducted with two to four factors. The optimal factor structure was three factors, with a Cronbach’s alpha coefficient and Cohen’s weighted kappa coefficient of 0.85 and 0.76, respectively. The intraclass correlation coefficient (ICC) of the test–retest was 0.93, the SEM was 0.72, and the SDC was 2.00. The model fit was good for the ARMT-J, with a three-factor structure.

## 1. Introduction

According to the World Health Organization (WHO), road traffic accidents kill approximately 1.3 million people worldwide and cause 20–50 million non-fatal injuries [1]. Many traffic accidents involve motor vehicle crashes, which are among the most critical public health problems worldwide. Aging is a factor that increases the risk of accidents. Ouchi et al. define the elderly as “pre-old age” at over 65 and under 75 and as “old age” at over 75 [2]. In contrast, other studies have shown that women may develop problems with activities of daily living (ADLs) and instrumental activities of daily living (IADLs) starting in midlife (age 55 and older), and some researchers have the problem of a focus on the prevalence of older adults (age 65 and older) [3]. Thus, the age range of aging may be interpreted differently in different studies. Older drivers are at increased risk for dementia and brain disease [4,5] and may experience sensory, cognitive, and physical impairments. Thus, the risk of accidents increases and driving skills are affected [6,7,8].

In Japan, the number of licensed drivers over 75 years of age is increasing, leading to many fatal accidents [9]; therefore, the Japanese government is promoting the creation of an environment that makes it easier for drivers to return their driver’s licenses, including aptitude counseling and support measures for older adult drivers considering discontinuing driving. Many older drivers may also cope by driving less frequently, limiting their driving distance, or increasing their non-driving opportunities [10,11]. However, the reduction or cessation of driving frequency increases the risk of depressive symptoms, long-term nursing home placement, and death among older drivers [12,13,14].

Furthermore, older adults’ cessation of driving is related to their physical frailty [15] and affects their level of social isolation [16], life satisfaction or time spent outside [17], inactivity and lack of interest [18], networking with friends [19], paid work, and volunteer activities [20]. Some studies have also shown that the factors influencing driving cessation in the elderly depend on personal experience and environment, such as hospitalization [21], area of residence [22,23], relationship with a partner [24], and financial situation [25]. Furthermore, the 10-year mortality rate for driving cessation is higher for men than women [12]. Thus, driving cessation in the elderly can cause various health-related problems. However, this risk could decrease if mobility can persist through public means or car use after driving cessation. Therefore, the importance of communication and advanced planning in driver-assistance interventions for the elderly has warranted attention [26]. Cessation of driving may improve some health outcomes among older adults, especially those who anticipate and prepare for it [27]. A study of older drivers aged 65~79 reported that participants who had considered discontinuing driving at the beginning of the study were more likely to maintain their quality of life and use public transportation after discontinuing driving regardless of age or health status compared to those who did not [28]. This perspective of preparing for driving cessation due to old age is essential because it facilitates the availability of services and programs to maintain life after driving cessation [29].

The Assessment of Readiness for Mobility Transition (ARMT) measures emotional and attitudinal readiness to cope with change and loss of mobility [30]. The original manual can be downloaded from the web [31]. The ARMT has never been translated into any foreign language except for the original version. Therefore, we obtained permission from Meuser et al. in 2020 and verified the linguistic validity of the Japanese version in 2022 [32]. However, reliability and validity, other than linguistic validity, have not been confirmed to date. This study aimed to confirm the reliability and validity of the Japanese version of the ARMT (ARMT-J).

## 2. Materials and Methods

### 2.1. Study Design and Participants

This was a cross-sectional test development study that used a self-administered questionnaire. We prepared a question–response table when there was difficulty understanding the questionnaire. An occupational therapist was also available to explain the same to the study participants. The study period was from December 2020 to March 2022. The original Consensus-based Standards for the selection of health Measurement Instruments (COSMIN) user manual specifies that the sample size for factor analysis (FA) is “7 times the number of items and ≥100” as the “very good” criterion [33]. Therefore, we set the sample size to 24 items × 7 = 168 items, the number of items in the original ARMT required to perform FA. The participants were patients who underwent rehabilitation at four hospitals in four cities of Japan (two in rural and two in urban areas), hospital employees, and their family members with driving licenses. Patients undergoing rehabilitation included those admitted to the hospital. Therefore, inpatients underwent their first ARMT when it was confirmed that they would resume driving after the discharge from the hospital. In the ARMT test–retest reliability test, we asked all participants if they could retest, and only those who agreed did so. The interval of retest is essential to prevent recall bias and to have little change in the measured items [33]. In addition, a test–retest study of health-related quality of life showed no significant difference between the two time intervals in a two-day or two-week test–retest study [34]. Based on these studies, we retested the participants for two days to two weeks after the initial assessment. Exclusion criteria were (1) those who could not understand the questionnaire due to aphasia or dementia, (2) those who had never held a driver’s license, and (3) those under 55 years old.

### 2.2. Measures

#### 2.2.1. Basic Information

Participants’ gender, age, education, disease, driving status, family members living with them, residential area, and accessibility of public transportation were identified using a questionnaire. The presence or absence of family members living with the participant was used as a nominal scale, and a five-point Likert scale was used for the residential area (1 urban to 5 rural), and convenience of public transportation (1 convenient to 5 inconvenient).

#### 2.2.2. Instruments

ARMT

It is an assessment sheet for counseling older adults with mobility transitions. This scale measures older adults’ emotional and attitudinal readiness for mobility transitions, and each item is answered on a five-point Likert scale from 1 (strongly disagree) to 5 (strongly agree). It consists of twenty-four items (or eight items in short form) and can measure four subscales: anticipatory anxiety (AA), perceived burden, (PB) avoidance (Av), and adverse situation (AS). The total score (TS) for the ARMT (range: 24–120) and its subscales (range: 1–5) consists of the sum of the item scores; the higher the score, the less prepared the patient is for the mobility transition.

The12-item short-form health survey: SF-12

The SF-12 is a rating scale for assessing health-related quality of life [35]. In SF-12, we used a three-component model validated to better fit the Asian population [36] and obtained scores for the physical component summary (PCS), mental component summary (MCS), and role/social component summary (RCS). It is scored on a scale of 0 to 100, with higher scores representing better health. In this study, the Cronbach’s alpha was 0.75.

Life Space Assessment: LSA

The LSA is a rating scale developed to measure the mobility status of the older adults [37]. This study used the Japanese translation of the LSA [38]. The study participants reported their mobility status four weeks before the evaluation. Life space consists of six levels of living space (0: mobility within the bedroom; 1: rooms inside the home besides the bedroom; 2: area outside the house; 3: neighborhood; 4: town or city lived in; 5: outside of town or city lived in). For each level, participants were asked (a) if they went to this level in the past four weeks; (b) if so, how often; (c) if they needed assistive devices or special equipment to reach that level; and (d) if they needed personal help to reach that level. The LSA score ranged from 0–120 points, with higher scores indicating better mobility. In this study, the Cronbach’s alpha was 0.66.

Geriatric Depression Scale—Short Version—Japanese (GDS-S-J)

Depression was measured using the GDS-S-J [39], which was developed from a shortened version [40] of the GDS [41] developed for screening depression among older adults. The GDS-S-J is a selection of 15 out of the 30 items in the original GDS. Participants answered “yes” or “no” to each item. Scores ranged from 0–15, with six or more points indicating depression. In this study, the Cronbach’s alpha was 0.85.

### 2.3. Statistical Analysis

Statistical analysis was performed using IBM SPSS (IBM, Armonk, NY, USA) Statistics for Windows, version 28, IBM SPSS AMOS for Windows, version 28, and R version 4.1.1, with a significance level of 5%. The basic information was used to confirm the proportion of each variable. Furthermore, we considered a floor or ceiling effect when the number of participants who fell into the minimum and maximum values exceeded 15% for the TS of the ARMT and that of each subscale (AA, Av, PB, and AS) and item, respectively. Finally, the Cronbach’s alpha coefficient was calculated to assess the internal consistency of the subscales [42].

### 2.4. Validity and Reliability

The COSMIN checklist contains items that cover: (1) validity (content validity, construct validity, and criterion validity), (2) reliability (internal consistency, reliability, and measurement error), (3) responsiveness, and (4) interpretability [33]. In this study, no intervention studies were conducted. Therefore, the responsiveness and interpretability were not examined. After validating the scale, we examined the reliability of the best-performing model.

#### 2.4.1. Validity

Content Validity:

Content validity is the degree to which a questionnaire’s content adequately reflects the constructs to be measured and should be assessed by determining the relevance and comprehensiveness of the items [33]. In the development of the original version of the ARMT, surface and content validity were confirmed using mixed study methods in a diverse sample of community-dwelling older adults [29,30,43]. To prepare the ARMT-J, we first checked the original authors and obtained permission to translate the text. Then, we translated the ARMT according to the International Society for Pharmacoeconomics and Outcomes Research translation guidelines for the Patient-Reported Outcomes measure (PROM) [44]. To confirm the understandability of the Japanese version, cognitive debriefing was conducted with five older adult persons aged 65 years or older living in the community [32].

Criterion Validity

Criterion validity refers to the degree to which the PROM scores are an appropriate reflection of the “gold standard” [33]. There was no standard evaluation index for comparison, and criterion validity was not performed in the original version [30]. For this reason, we did not conduct hypothesis testing in this study but confirmed the correlations with other related rating scales in construct validity.

Construct Validity

Cross-cultural validity: This item aims to ascertain the extent to which the performance of the translated or culturally adapted questionnaire items adequately reflects the performance of the items in the original version of the PROM [33]. We used confirmatory factor analysis (CFA) to examine whether a four-factor structure similar to the original ARMT could result from an aged sample similar to the original version. The χ^2^ value is more likely to be significant when the sample size is large for assessing the model’s goodness of fit [45]. However, these were also checked because lower values of χ^2^/df and AIC are considered a better fit [46]. Model stability was also confirmed by the goodness of fit index (GFI > 0.90), normed fit index (NFI > 0.90), comparative fit index (CFI > 0.90), Tucker–Lewis index (TLI > 0.90), and root mean square error of approximation (RMSEA < 0.080) [47,48].

Structural validity: This item checks the extent to which the PROM scores adequately reflect the measured construct’s characteristics [33]. The exploratory factor analysis (EFA) of the original version of the ARMT yielded five factors from a total of 24 items. However, factors 4 and 5 lacked internal reliability (α = 0.54 and 0.44, respectively), so factors 4 and 5 were finally merged and constructed as four factors [30]. When the cross-cultural validity of this study failed to fit a factor structure model similar to that of the original version, Kaiser–Meyer–Olkin (KMO) and Bartlett’s sphericity tests were used to evaluate the adequacy and suitability of the sample of the research before performing the factor analysis [49]. Further, Velicer’s minimum average partial (MAP) test [50] and parallel analysis [51] were used to examine the number of valid factors, followed by EFA. We first used Shapiro–Wilk and Mardia’s kurtosis tests to confirm univariate and multivariate normality. When data normality could not be obtained, the principal factor method was used [52]. The factor rotation used the direct oblimin method, which is an oblique rotation, assuming that the factors after the EFA are related [45]. The cut-off for a factor should be greater than 0.40. Each item should load less than 0.30 on the other factors and demonstrate a difference of 0.20 between their primary and alternative factor loading [45]. All factor models obtained by the MAP test and parallel analysis were subjected to repeated FA until these conditions were met. Factor models obtained by EFA were subjected to CFA to determine the model’s goodness of fit. For the best-fitting model, an attempt was made to improve the model’s goodness of fit by adding the covariance of the inter-item errors.

Hypotheses testing: Referring to the development study of the original version of the ARMT, the hypothesis was formulated using the SF-12, GDS-S-J, and LSA, standardized in the Japanese version, to obtain convergent validity of the ARMT-J.

SF-12: Three domains were utilized: PCS, MCS, and RCS. We hypothesized that if respondents had no physical and mental problems and believed they had a social role to play, this would indicate greater overall readiness to cope with the mobility transition, and a weak negative correlation between SF-12 scores and ARMT scores was hypothesized.

LSA: We hypothesized that those who were more prepared for life after the interruption of driving would be less likely to experience a narrower range of life and that a weak negative correlation between LSA and ARMT-T would occur.

GDS-S-J: Some of the participants in this study were hospitalized, and some were unable to resume driving. Therefore, we hypothesized that the GDS-S-J score would be lower for those not ready for transition to mobility and that there would be a weak negative correlation between the GDS-S-J score and AMRT.

#### 2.4.2. Reliability

Internal Consistency

Internal consistency is the degree of interrelationship between the items [32]. It was calculated using the reliability coefficient Cronbach’s α, with 0.61–0.70 being acceptable, 0.71–0.80 being good and acceptable, and 0.81–0.90 being good [53].

Reliability (Test–Retest Reliability)

Reliability checks the extent to which the score of an unchanged patient is the same when measured repeatedly under several conditions [33]. In this study, we confirmed the test–retest reliability when ARMT evaluations occurred at a specific time interval. The participants were those who participated in the first study and gave consent to retest the participants of the study. Statistical analysis consisted of the intraclass correlation coefficient (ICC) [54] and Cohen’s weighted kappa coefficient [42,55]. In addition, Cronbach’s alpha coefficient was used for each ARMT factor and internal consistency of the TS [32,42]. The measuring instrument has moderate reliability when the ICC is between 0.5 and 0.75, good reliability when the ICC is between 0.75 and 0.9, and excellent reliability when it exceeds 0.9 [56]. Cohen’s weighted kappa coefficient (k) was set to poor (k < 0.4), fair to good (0.4 ≤ k < 0.75), and excellent (k < 0.75) according to Fleiss’s criteria for agreement of kappa values [57].

Measurement Error

Standard error of measurement (SEM) indicates whether the change in scores is an actual change, and the minor detectable change indicates a slight intra-individual change in scores; $\mathrm{SEM}=\mathrm{SD}\left(\mathrm{Standard}\;\mathrm{Deviation}\right)\times \sqrt{1}-\mathrm{ICC}$. SDC (smallest detectable change) was calculated based on SEM, $\mathrm{SDC}=1.96\times \mathrm{SEM}\times \sqrt{2}$ [42].

## 3. Results

### 3.1. Participants and Basic Information

We recruited 212 participants with written study consent and excluded 37 participants who met the exclusion criteria and one whose ARMT and SF36 were not listed. The investigation and subsequent analysis involved 173 participants. For the retest, 128 participants agreed, and all answers were analyzed. The mean (SD) age was 66.8 (8.5). Overall, 33% of the participants were women. In addition, 45% were still driving, and 55% were in driving cessation. Rural and urban residents accounted for 66% and 34%, respectively. Table 1 presents other basic information. No ceiling or floor effects occurred in the TS and subscales of the ARMT-J. For each item, the ceiling effect was found in ARMT-J1, 3, 4, 6, 9, 13, and 19 and the floor effect in ARMT-J7, 8, 14, 15, 17, 21, 22, 23, and 24. For the other items, both the floor and ceiling effects were confirmed. Cronbach’s alpha coefficients for the ARMT-J sub-factor items based on the original factor structure were AA = 0.89, PB = 0.80, Av = 0.44, and AS = 0.59.

### 3.2. Validity: Construct Validity

#### 3.2.1. Cross-Cultural Validity

In the CFA, as in the original version, χ^2^ = 499.071, *p* < 0.0001, and χ^2^/df = 2.09, and except for RMSEA (=0.077), none of the fit indices showed acceptance criteria: GFI = 0.793, NFI = 0.737, CFI = 0.844, TLI = 0.825, and AIC = 607.012, and the compliance rate was low.

#### 3.2.2. Structural Validity

The KMO value was 0.903, and the χ^2^ value by Bartlett’s test of sphericity was 1799.822 (*p* < 0.001), indicating a reasonable value for the factor analysis. The Shapiro–Wilk test of univariate normality showed that the distribution deviated significantly from the normal distribution for all variables (*p* < 0.001), and Mardia’s kurtosis test did not establish multivariate normality for the item responses (skewness, 3570.48.215; kurtosis, 10.7; *p* < 0.001). In addition, there were six factors for the parallel test and two factors for the MAP test. The EFA was performed by direct oblimin using principal factor analysis, but the solutions did not converge for factors 5 and 6. Therefore, the EFA was performed using factors 2–4. (Table 2).

EFA: In the two-factor model, the first factor consisted of numbers1, 3, 10, 12, 14–16, 18, 21, and 22 of the original ARMT, which included eight AA items and one item each for AS and PB. The second factor consisted of numbers 5, 7, and 8 of the original ARMT, which included two items for AS and one for Av. In the three-factor model, the first factor consisted of the original ARMT numbers 14–16 and 21 and all AA items. Factor 2 consisted of the original ARMT items 5, 7, and 8, similar to those in the two-factor model. Finally, the third factor was the original ARMT numbers 2, 3, 9, and 24, which contained one AA item and three PB items. In the four-factor model, factor 1 was numbers 10, 12, 14–16, and 21 of the original ARMT, and all items were AA. Factor 2 was numbers 5, 7, and 8 of the ARMT, similar to the other factor models. Factor 3 was numbers 6 and 9 of the original ARMT, all PB items, and factor 4 was numbers 1, 2, and 4 of the ARMT, a mixture of all items except AS.

CFA: CFA considers each factor structure model obtained by EFA. Table 3 shows the results of the model fit using CFA. The three-factor model had the best fit by CFA, with the following factor names according to the original version: (1) AA; (2) AS; and (3) PB. The final three-factor model was improved by adding covariance of item-to-item errors, as shown in Figure 1.

#### 3.2.3. Hypotheses-Testing

Table 4 shows the correlation matrix between the ARMT TS and subscale and other scales. As hypothesized, there was a weak positive correlation between TS and GDS in ARMT (r = 0.324), and no correlation with ARMT was obtained for all items in the LSA and SF12.

### 3.3. Reliability

#### 3.3.1. Internal Consistency

Table 5 shows the mean, SD, median, and the interquartile range, Cronbach’s alpha coefficient, Cohen’s weighted κ, ICC, SEM, and SDC for the ARMT-J and its subscales and each item in the three-factor model. Cronbach’s alpha confirmed good internal consistency, acceptable for AS, good and acceptable for TS and PB, and good for AA. In addition, Cohen’s weighted κ was excellent for the TS and fair to good for the other subscales and items. Finally, the reliability of each item was confirmed to be 0.54–0.84: moderate to good (Table 5).

#### 3.3.2. Measurement Error

The SEM of the ARMT TS was 0.72, SDC = 2.00. Thus, a change of two or more points from the initial ARMT score to the reassessment for each factor was considered a true change (Table 5).

## 4. Discussion

This study aimed to confirm the reliability and validity of the ARMT-J. The participants were the same as in the original version and were aged 55 years or older, with a mean of 66.8 years (SD = 8.5), which was slightly younger than the original sample of 71 years [30]; however, we believe that the age sample was generally similar. Reliability and validity studies revealed that the factor structure of the ARMT-J differs from that of the original version in that it has a three-factor structure. The ARMT-J and its subscales in the three-factor structure showed high internal consistency and moderate test and retest reliability. In addition, correlations with correlated health-related scales indicated that the scale is theoretically stable.

### 4.1. Validity

First, a CFA for the cross-cultural analysis was conducted, assuming the ARMT to have the same four factors (24 items) as in the original version, but the model’s fit was poor. Therefore, MAP tests and parallel analyses were conducted, and the EFA assumed the number of factors to be 2 to 4. The CFA again confirmed the results obtained from the EFA, and although the set criteria were met for all factor structures, the three-factor model was the best.

The AA factor was a mixture of two PB items and one AS item from the original ARMT in a two-factor structure. However, the other factor structures consisted only of AA items, as in the original version. The concept of AA in the original version was “anxiety and felt concern about loss of personal integrity and independence in the face of significant mobility loss”. Therefore, the factor was named “anticipatory anxiety.” In the two-factor structure, the number of factors was small, so other items in the original version were mixed. However, in the three- and four-factor structures, although some items were deleted, there was no mixing of other items, and we believe that the factors in AA can be understood in the same way as in the original version.

The AS factor items consisted of similar items (PB, ARMT7,8, Av, ARMT5) in all factor structures, with only one item from the ARMT, namely Av, mixed in. The ARMT 5 entry, for this reason, is “I wish others would stop talking to me about my mobility”. ARMT 7 is “Other people simply do not understand what it’s like to have limited mobility”. ARMT 8 is “It is devastating for older people to have someone take away their car keys.”

These items can be understood as items related to others’ understanding of traffic vulnerability. In the original version, the concept of AS is “A general perception of significant mobility loss as very harmful to individual well-being and quality of life”. The factor is named “adverse situation.” In the social framework, being evaluated unfavorably by others can be understood as “adverse situation” (AS) in the same way as in the original version.

The PB items were confirmed in three- and four-factor structures. In the three-factor structure, only one of the four items in ARMT2 contained items from the original AA, whereas all two items in the four-factor structure consisted of the same items as in the original PB. In the original version, the concept of PB is “worry associated with becoming overly dependent and a burden on others” and is named as “perceived burden.” ARMT2, mixed in the three-factor structure, is “asking others for help with mobility means that I am losing my independence”. The word “other” is present in the sentence. This suggests that respondents may have viewed the ARMT2 items as items related to the burden on others, and even if this item is mixed, we believe that it can be understood as “perceived burden” (PB) as in the original version.

Finally, with regard to the Av factor, the original version defines it as “a general resistance to address the topic of mobility loss” and “avoidance.” The only items with mixed Av items were ARMT5 in the all-factor model and F4 (ARMT4) in the four-factor structure, and no factor structure similar to Av in the original version was identified. In the four-factor structure, three items selected as F4 were ARMT1 (AS), ARMT2 (PB), and ARMT4 (Av), but all were factor items that differed from the original version and could not provide the same understanding as the original version. The reasons for this are as follows: First, Cronbach’s alpha coefficient for Av −0.62 in the original version was low, which means that the internal consistency was low. Second, the sample size was smaller than that of the original version, and the sample characteristics may have been affected by factors such as a smaller number of women and a more significant number of non-drivers in the sample. Based on the results of this study, the three-factor structure appeared to be the best model for Japanese adults aged 55 years or older. The number of factors and items in the original version has decreased. Careful interpretation of these factors is required when using this model. CFA should also be conducted in the original version, and it is necessary to consider whether Av items should be excluded.

In hypothesis testing, only the GDS showed weak-to-moderate correlations, as hypothesized, while the other hypotheses were not supported or not significantly different. Furthermore, ARMT and SF12 were generally consistent with a slight negative correlation although they were not significantly different in the MCS. The RCS did not examine the original version, but as hypothesized, it was negatively correlated; that is, the less ready they were for mobility transition, the lower their social role scores were. Only the PCS score showed a slightly positive but not significant correlation. This may be because some participants took part in the study while being hospitalized. Many were unprepared for the transition of mobility, including driving interruptions following a sudden event of illness or injury. Therefore, contrary to our hypothesis, those with higher physical function would not be aware of interrupting driving, which may have led to this result. The LSA assumed negative and weak correlations; however, the results showed a mixture of positive and negative correlations, and almost no correlations were identified. The reason may be that many participants were hospitalized, and even if they could go outdoors, they would only do so within the premises, so there was no difference in the scores. In the future, it will be necessary to improve the validity of the data by adjusting the target to the original version and changing the outcome when hospitalized patients are included.

### 4.2. Reliability

In the internal consistency results, only AS showed a slightly low value of 0.67, which was acceptable, while TS and AA were good at 0.85 and 0.87, respectively, and PB was good and acceptable with 0.73, which was good reliability. This result is consistent with those of previous studies [30], and we believe that the overall stability was good. The test-retest results showed good results for TS with Cohen’s weighted κ at 0.76 (0.71–0.08) and ICC at 0.93 (0.91–0.95). The subscales also showed good results, and each item was confirmed to have moderate or high reliability (Table 5). Although hospitalized patients participated in the evaluation, they did so while it was clear whether they would be able to resume driving. Therefore, the results are stable. A previous study [30] citing an unpublished paper also showed a good correlation of 0.84 (*p* < 0.01) for test–retest TS. Our study supported the results of a previous study. SDC is a criterion for determining whether the observed change is an actual change beyond the measurement error [42]. If the change exceeds the SDC, it indicates that a “true change” has occurred in that individual. Therefore, it is recommended that changes in ARMT-J TS confirm the change and use SDC as a reference value when evaluating the effect of the intervention in the future.

### 4.3. Limitation

This study has several limitations. First, the sample lacked proper representation. The participants were patients, staff, and their families admitted to a hospital in Japan, and the survey was conducted in urban and rural areas; however, the sample was 66% urban and biased. In the original version, women accounted for 78% of the sample; however, in the present study, men accounted for 67%, and most of the population in the sample were hospitalized or outpatients. Since there are some differences from the original version, additional research is needed in the future based on adjustments of the sample characteristics.

Second, some psychometric properties such as responsiveness and interpretability were not examined. In the future, we believe that it is necessary to link these findings to cohort and intervention studies to increase their reliability.

Finally, the factor structure differed from that of the original version. CFA did not occur in the original version, and ARMT was created using only the EFA. In this study, the three-factor structure was the most stable result, and all factors except Av can generally be used with the same meaning as in the original version, but some items were deleted in AA, and some items were mixed in PB and AS, which were different from the original. Therefore, when using the original version, the user should consider that (1) the items in the factors are slightly different from those in the original version, and (2) the factor structure is smaller than that in the original version.

### 4.4. Implication

The original version of the ARMT is an index created through qualitative research using a wide variety of samples for use in counseling [43]. However, the reliability and validity of the original ARMT have never shown sufficient confirmation with classical test theory. The ARMT-J used in this study was developed after checking the original authors’ contents. Therefore, the ARMT-J with the three-factor model developed in this study should be used as an effectiveness index in interventional studies. However, when the ARMT-J is used for counseling purposes, we believe that using the 24-item version of the ARMT-J, in which no items have been deleted, will provide a better understanding of the mobility transition readiness of people who have discontinued driving. The findings suggest that ARMT-J may be helpful for education and research.

## 5. Conclusions

This study examined the reliability and validity of the ARMT-J. Cross-cultural validity, structural validity, and hypotheses testing were verified for construct validity. In addition, the internal consistency, test–retest reliability, and measurement error were verified to be reliable. Consequently, the three-factor structure proved to be the best factor. The factor structures were AA, PB, and AS except for Av in the ARMT-J-14. However, as the factor structure differs from the original ARMT, care must be taken when interpreting the results. Therefore, in the future, it will be necessary to examine the interpretability of the results while using ARMT in parallel with the original version.

## Figures and Tables

**Figure 1 ijerph-19-13957-f001:**
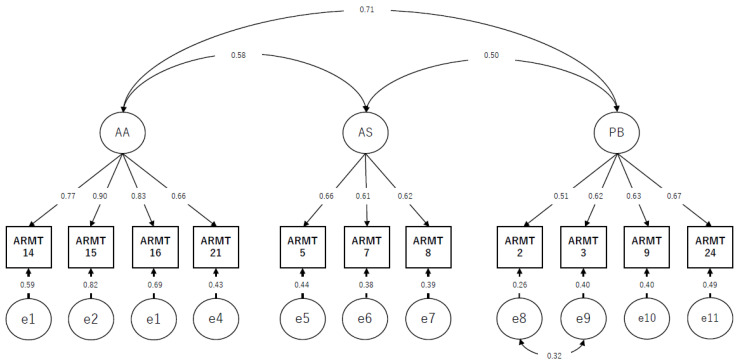
Results of confirmatory factor analysis with standardized estimates and error covariances for the Japanese version of the Assessment of readiness for mobility transition (*n* = 173). Abbreviations: AA; Anticipatory Anxiety, AS; Adverse Situation, PB; Perceived Burden. Note: Model fit indices χ^2^ = 59.289, P = 0.025, χ^2^/df = 1.482, GFI = 0.94, NFI = 0.92, CFI = 0.97, TLI = 0.96, RMSEA = 0.053.

**Table 1 ijerph-19-13957-t001:** Demographic characteristics of participants (*n* = 173).

		Number (%)
Sex	Female	57 (33)
	Male	116 (67)
age	55–60	50 (29)
	61–65	35 (20)
	66–70	32 (19)
	71–75	28 (16)
	≥76	28 (16)
Type of Living	Alone	38 (22)
	With someone	135 (78)
Education	Junior high school	18 (10)
	High school	76 (44)
	Vocational school	42 (24)
	University	34 (20)
	Graduate school	3 (2)
Primary disease	None (Wellness)	30 (17)
	Stroke	95 (55)
	Traumatic brain injury	6 (4)
	Bone fracture	9 (5)
	Spinal cord injury	15 (9)
	Others	18 (10)
Driving status	Driving	77 (45)
	Driving cessation	96 (55)
Residential area	Rural	115 (66)
	Urban	58 (34)
Accessibility of public transportation	Exclent	34 (20)
	Very good	68 (39)
	Fair	39 (22)
	Poor	22 (13)
	Unacceptable	10 (6)

**Table 2 ijerph-19-13957-t002:** EFA of the Japanese version of the ARMT (*n* = 173, principal axis factoring with oblimin rotation).

Items	Communality	2 Factors		3 Factors			4 Factors			
		F1	F2	F1	F2	F3	F1	F2	F3	F4
ARMT1	0.46	**0.409**	0.187				0.107	0.066	−0.045	**−0.657**
ARMT2	0.51			−0.011	0.056	**0.604**	0.111	−0.082	0.192	**−0.511**
ARMT3	0.48	**0.456**	0.049	−0.071	−0.048	**0.859**				
ARMT4	0.48						0.013	0.092	0.004	**−0.646**
ARMT5	0.46	0.059	**0.694**	0.001	**0.620**	0.099	−0.042	**0.600**	0.027	−0.251
ARMT6	0.52						−0.102	0.104	**0.737**	−0.094
ARMT7	0.37	−0.015	**0.647**	−0.105	**0.702**	0.059	−0.045	**0.629**	0.116	−0.033
ARMT8	0.33	0.176	**0.424**	0.217	**0.520**	−0.114	0.247	**0.509**	−0.045	0.084
ARMT9	0.42			0.162	0.009	**0.455**	0.189	−0.039	**0.632**	0.064
ARMT10	0.57	**0.743**	−0.017				**0.575**	−0.033	0.066	−0.264
ARMT11	0.33									
ARMT12	0.49	**0.611**	0.045				**0.497**	0.090	0.123	−0.061
ARMT13	0.13									
ARMT14	0.63	**0.685**	0.114	**0.596**	0.262	0.040	**0.683**	0.240	0.089	0.091
ARMT15	0.73	**0.945**	−0.120	**0.822**	0.010	0.121	**0.846**	−0.007	0.072	−0.025
ARMT16	0.71	**0.985**	−0.261	**0.910**	−0.119	0.039	**0.879**	−0.097	−0.026	−0.058
ARMT17	0.46									
ARMT18	0.43	**0.448**	0.122							
ARMT19	0.28									
ARMT20	0.29									
ARMT21	0.56	**0.596**	0.160	**0.451**	0.221	0.135	**0.482**	0.159	−0.036	−0.210
ARMT22	0.62									
ARMT23	0.54	**0.531**	0.226							
ARMT24	0.48			0.172	0.060	**0.463**				
Factor correlation		F1	F2	F1	F2	F3	F1	F2	F3	F4
F2		0.558	-	0.469	-		0.414	-		
F3				0.534	0.383	-	0.419	0.287	-	
F4							−0.478	−0.366	−0.384	-

Note: The Bold type in the table indicates selected factor items. The following rules were applied to the factor cutoffs. Namely, “It is recommended that satisfactory variables (a) load onto their primary factor above 0.40, (b) load onto alternative factors below 0.30, and (c) demonstrate a difference of 0.20 between their primary and alternative factor loadings.” Abbreviations: ARMT, assessment of readiness for mobility transition; EEA, exploratory factor analysis, F, factor.

**Table 3 ijerph-19-13957-t003:** CFA of the Japanese version of the ARMT.

Model	Items	_χ_2	df	χ^2^/df	*p*	GFI	CFI	RMSEA	AIC
Two-factor model	F1: 1, 3, 10, 12, 14–16, 18, 21, 23	120.19	64	1.88	*p* < 0.000	0.903	0.938	0.071	174.19
	F2: 5, 7, 8								
Three-factor model	F1: 14–16, 21	70.67	41	1.72	*p* < 0.003	0.930	0.956	0.065	120.67
	F2: 5, 7, 8								
	F3: 2, 3, 9, 24								
Four-factor model	F1: 10, 12, 14–16	137.21	71	1.93	*p* < 0.000	0.901	0.931	0.074	205.21
	F2: 5, 7, 8								
	F3: 6, 9								
	F4: 1, 2, 4								

Note: Abbreviations: df, degrees of freedom; CFA, confirmatory factor analysis; GFI, goodness of fit index; CFI, comparative fit index; RMSEA, root means square error of approximation; AIC, Akaike’s information criterion.

**Table 4 ijerph-19-13957-t004:** Correlation coefficients between ARMT-J, GDS, LSA, and SF-36.

	ARMT-AA	ARMT-PB	ARMT-AS	ARMT-TS	GDS	LSA	SF-36 PCS	SF-36 MCS	SF-36 RCS
ARMT-AA	1.000								
ARMT-PB	0.566 **	1.000							
ARMT-AS	0.492 **	0.334 **	1.000						
ARMT-TS	0.883 **	0.806 **	0.69 **	1.000					
GDS	0.250 **	0.262 **	0.238 **	0.324 **	1.000				
LSA	0.033	0.073	−0.126	0.016	−0.163 *	1.000			
SF-36 PCS	0.092	0.011	0.044	0.061	−0.054	0.302 **	1.000		
SF-36 MCS	0.005	−0.076	−0.005	−0.043	−0.42 **	−0.118	−0.113	1.000	
SF-36 RCS	−0.173 *	−0.120	−0.059	−0.165 *	−0.266 **	0.23 **	−0.239 **	−0.095	1.000

Note: Abbreviations: TS, total score; LSA, Life Space Assessment; GDS-S-J; Geriatric Depression Scale-Short Version-Japanese; SF-12, The12-item short-form health survey; PCS, physical component summary; MCS, mental component summary; RCS, role/social component summary. Spearman’s rank-order correlation analysis calculated correlation coefficients. ** *p* < 0.01; * *p* < 0.05.

**Table 5 ijerph-19-13957-t005:** Test–retest reliability, standard of measurement error, and smallest detectable change of the ARMT-J.

		Test		Retest					
Scale	Score Range	Mean (SD)	Median (IQR)	Mean (SD)	Median (IQR)	Cohen’s Weighted κ (95% CI)	ICC (95% CI)	SEM	SDC
ARMT-TS	11–55	31.91 (10.29)	31 (24.5–41)	31.94 (10.81)	31 (24–39.5)	0.76 (0.71–0.80)	0.93 (0.91–0.95)	0.72	2.00
ARMT—PB	4–20	12.54 (4.33)	13 (9–16)	12.49 (4.44)	12 (9–16)	0.65 (0.58–0.72)	0.84 (0.78–0.88)	0.31	0.86
Item 2	1–5	3.01 (1.42)	3 (2–4)	3.01 (1.34)	3 (2–4)	0.48 (0.37–0.58)	0.60 (0.48–0.70)	0.11	0.30
Item 3	1–5	3.20 (1.33)	3 (2–4)	3.20 (1.36)	3 (2–4)	0.60 (0.50–0.70)	0.72 (0.62–0.79)	0.10	0.29
Item 9	1–5	3.27 (1.36)	4 (2–4)	3.15 (1.43)	3 (2–4)	0.70 (0.61–0.78)	0.81 (0.74–0.86)	0.11	0.28
Item 24	1–5	3.06 (1.44)	3 (2–4)	3.12 (1.33)	3 (2–4)	0.57 (0.47–0.67)	0.69 (0.59–0.77)	0.10	0.28
ARMT—AS	3–15	8.51 (3.23)	9 (6–11)	8.45 (3.19)	8 (6–11)	0.73 (0.60–0.74)	0.85 (0.79–0.89)	0.24	0.66
Item 5	1–5	2.92 (1.48)	3 (1–4)	2.92 (1.42)	3 (2–4)	0.58 (0.47–0.68)	0.69 (0.58–0.78)	0.10	0.30
Item 7	1–5	2.88 (1.28)	3 (2–4)	2.97 (1.21)	3 (2–4)	0.45 (0.33–0.57)	0.54 (0.40–0.65)	0.10	0.27
Item 8	1–5	2.71 (1.44)	3 (1–4)	2.55 (1.38)	2 (1–4)	0.65 (0.56–0.74)	0.77 (0.69–0.83)	0.10	0.28
ARMT—AA	4–20	10.86 (4.82)	11 (7–14.5)	11.00 (4.67)	11 (8–14)	0.68 (0.60–0.75)	0.84 (0.78–0.89)	0.35	0.96
Item14	1–5	2.61 (1.39)	2 (1–4)	2.71 (1.40)	3 (1–4)	0.63 (0.55–0.73)	0.76 (0.68–0.83)	0.10	0.27
Item15	1–5	2.69 (1.44)	3 (1–4)	2.70 (1.39)	3 (1–4)	0.60 (0.51–0.69)	0.76 (0.67–0.82)	0.10	0.29
Item16	1–5	2.98 (1.46)	3 (2–4)	2.96 (1.39)	3 (2–4)	0.57 (0.47–0.68)	0.66 (0.55–0.75)	0.11	0.30
Item21	1–5	2.57 (1.31)	2 (1–2)	2.61 (1.24)	2 (2–4)	0.64 (0.55–0.74)	0.74 (0.66–0.81)	0.10	0.28

Note: Abbreviations: SD, standard deviation; ICC, intra-class correlation coefficient; SEM, standard error of measurement; SDC, smallest detectable change. Test–retest survey (*n* = 124).

## Data Availability

Data were not publicly deposited. However, we can share data and analyses that underpin the findings reported in this study. These are available on request from sa-nasu@janrc.or.jp.

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
