# Peer review of "Reliability and Validity of the Japanese Version of the Assessment of Readiness for Mobility Transition (ARMT-J) for Japanese Elderly"

_ijerph, 2022, doi:10.3390/ijerph192113957_

Round 1
Reviewer 1 Report
Thanks to authors for representing the study about developing the Japanese Version of the Assessment of Readiness for Mobility Transition (ARMT-J). Driving is an important mean of transportation and the changes over course of ageing can lead to situations where the driving ability reduces substantially or driving could not be performed anymore.
The authors have planed study based on the recommendations of COSMIN user manual and therefore the study could serve as the good example of measurement adaptation.
I have a few comments regarding the presentation of methods and results, related to the sample description. Somehow I got confused about the sample description and this could be improved in the manuscript.
1) In Abstract indicated that "...ARMT-J was administered to 174 patients and staff members undergoing rehabilitation at hospitals in Japan..." (lines 17-18) but in section 2.1. more precise description of participants background is given (lines 67-69). In results, section 3.1. more precise information on participants number is given (n=173). I would recommend to give precise information on sample already in Abstract.
2) In section 3.1. first sentence about participants is confusing (line 228). Does there were 212 potential participants and only 173 included in the study? Explanation is given only for one person with missing ARMT and SF36 responses.
3) In section 2.1. is written that "...The sample size consisted of 24 items × 7 = 168 cases..." (lines (65-66) but idea is not clear to me. The number of items (n=24) refer to the ARMT but I idea how does it leads to 168 cases is not clear to me.
4) Table No1 represents the demographic characteristics of participants. The last characteristic refer to "Useful of public transportation" - does it represent participants ability to use the public transport?
Reviewer 2 Report
Dear researcher(s), you are addressing an important and meaningful gap. Your paper is well-written and it has some important results, and if you edit your paper it can be much more effective. Here are some humble suggestions to improve the paper, I would do the following to strengthen the paper. I have enjoyed reading the paper and am looking forward to seeing the paper published. You could increase the effect of your paper with some more recent studies suggested below or any other studies and not use the suggested ones.
